# Antibiotic susceptibility profiles of ocular and nasal flora in patients undergoing cataract surgery in Taiwan: an observational and cross-sectional study

Yun-Hsuan Lin,[1] Yu-Chuan Kang,[1] Chiun-Ho Hou,[1,2] Yhu-Chering Huang,[2,3] Chih-Jung Chen,[2,3] Jwu-Ching Shu,[4] Pang-Hsin Hsieh,[2,5] Ching-Hsi Hsiao[1,2]

[1]Department of Ophthalmology, Chang Gung Memorial Hospital, Keelung, Keelung, Taiwan
[2]College of Medicine, Chang Gung University, Kwei-Shan, Taiwan
[3]Division of Paediatric Infectious Diseases, Department of Pediatrics, Chang Gung Memorial Hospital, Keelung, Taiwan
[4]Department of Medical Biotechnology and Laboratory Science, College of Medicine, Chang Gung University, Kwei-Shan, Taiwan
[5]Department of Orthopaedic Surgery, Chang Gung Memorial Hospital, Kwei-shan, Taiwan

**Correspondence to**
Dr Ching-Hsi Hsiao;
hsiao.chinghsi@gmail.com

## ABSTRACT

**Objective** To investigate the conjunctival and nasal flora and the antibiotic susceptibility profiles of isolates from patients undergoing cataract surgery.

**Design** Observational and cross-sectional study.

**Setting** A single-centre study in Taiwan.

**Participants** 128 consecutive patients precataract surgery.

**Primary and secondary outcome measures methods** Conjunctival and nasal cultures were prospectively obtained from 128 patients on the day of cataract surgery before instillation of ophthalmic solutions in our hospital. Isolates and antibiotic susceptibility profiles were identified through standard microbiological techniques. Participants were asked to complete a questionnaire on healthcare-associated factors.

**Results** The positive culture rate from conjunctiva was 26.6%, yielding 84 isolates. Coagulase-negative Staphylococci were the most commonly isolated organisms (45.2%), and 35% of staphylococcal isolates were methicillin-resistant. Among staphylococcal isolates, all were susceptible to vancomycin, and 75%–82.5% were susceptible to fluoroquinolones. Methicillin-resistant isolates were significantly less susceptible than their methicillin-sensitive counterparts to tobramycin, the most commonly used prophylactic antibiotic in our hospital (28.6% vs 69.2%; p=0.005). The positive culture rate from nares for Staphylococcus aureus was 21.9%, and six isolates were methicillin-resistant. No subjects had S. aureus colonisation on conjunctiva and nares simultaneously. There were no associated risk factors for colonisation of methicillin-resistant Staphylococci.

**Conclusion** The most common conjunctival bacterial isolate of patients undergoing cataract surgery was coagulase-negative Staphylococci in Taiwan. Because of predominant antibiotic preferences and selective antibiotic pressures, Staphylococci were more susceptible to fluoroquinolones but less to tobramycin than in other reports. Additionally, methicillin-resistant Staphylococci exhibited co-resistance to tobramycin but not to fluoroquinolones.

### Strengths and limitations of this study

► This observational and cross-sectional study was to investigate the conjunctival and nasal flora and the antibiotic susceptibility profiles of isolates from patients undergoing cataract surgery in Taiwan where such information is limited.

► Our results delineate methicillin-resistant Staphylococci co-resistance to tobramycin but not to fluoroquinolones in Taiwan.

► Associated risk factors for methicillin-resistant Staphylococci colonisation were not found in current study.

► The culture rate from conjunctiva was lower than that obtained from other studies.

► Small sample sizes in the current study may limit statistical significance.

## INTRODUCTION

Cataract surgery is one of the most common surgical procedures performed by ophthalmologists. Endophthalmitis is a rare but devastating complication of cataract surgery. Gram-positive pathogens are responsible for 60%–80% of endophthalmitis; among them, coagulase-negative Staphylococci (CNS) are the most frequent isolates.[1–3] The source of pathogens for endophthalmitis is mainly the ocular surface of the patients. Additionally, nasal mucosa serves as a reservoir of organisms for the conjunctiva and nasolacrimal system. Speaker et al demonstrated that organisms isolated from the vitreous of endophthalmitis were genetically indistinguishable from those derived from the eyelids, conjunctiva and noses in 82% of cases.[4] Thus, evaluation of the flora and their drug susceptibility may serve as the success of empiric perioperative antibiotic use. However, the risk and success factors may vary across geographic regions

and over periods of time.[5–7] So, local epidemiological data must be established, and periodic surveys must be conducted to serve as references for clinicians.

Methicillin-resistant *Staphylococcus* sp are not only resistant to methicillin, but also to all β-lactam antibiotics. Additionally, methicillin resistance might also be related to resistance to other classes of antibiotics, which may limit treatment options. Methicillin-resistant *Staphylococcus* is a major concern for global public health. Recently, studies have focused on endophthalmitis caused by methicillin-resistant *Staphylococcus*.[8–11] Some notable features of methicillin-resistant *Staphylococcus*-associated endophthalmitis are as follows: a rise in the reported incidence rate,[8 10 11] an association with poor visual outcome[8 11] and an association with resistance to fluoroquinolones.[8 11] Methicillin-resistant *Staphylococcus aureus* (MRSA) colonisation is an important risk factor for subsequent MRSA- related infections in other fields.[12 13] Identifying patients with a high risk of developing methicillin-resistant *Staphylococcus* infections may be helpful in preventing severe complications from cataract surgery.

To effectively prescribe empiric antibiotics, a profile of likely pathogens in a local community is required. However, information regarding bacterial colonisation on ocular surfaces in Taiwan is limited. Additionally, methicillin-resistant *Staphylococcus* is prevalent in Taiwan. However, it is unknown whether these organisms colonise the ocular surface. Therefore, we conducted a study to investigate the conjunctival and nasal flora and the antibiotic susceptibility profiles of the isolates from patients undergoing cataract surgery at Chang Gung Memorial Hospital (CGMH). Furthermore, we focused on methicillin-resistant *Staphylococcus* sp and explored the associated risk factors. We intend to provide baseline data to help local clinicians choose appropriate antibiotic prophylaxes for their patients.

## METHODS
### Study participants and data collection
From 1 August 2012 to 31 July 2013, we enrolled patients >18 years old who were scheduled for elective cataract surgery. Exclusion criteria included pregnancy, use of oral or topical antibiotics within 3 months before surgery, and active ocular infection or conjunctivitis. The nature and procedures of the study were fully explained to all participants, and informed consent was obtained before recruitment. The Institutional Review Board of CGMH approved the study.

Participants were asked to complete a questionnaire on healthcare-associated factors. Questions regarding hospitalisation, surgery, dialysis, duration of stay in healthcare facilities, catheter use, use of immunosuppressive medications and antimicrobials, comorbid medical conditions (cardiovascular disease, liver disease, asthma, chronic obstructive pulmonary disease, renal failure, HIV, malignancy and diabetes mellitus), sociodemographic data (alcohol abuse, homelessness, close contact with children

<5 years and close contact with children who attend daycare) and ocular history (ocular disease, surgery, contact lens use and topical medication) were included.

### Sampling of bacteria
Prior to the preoperative instillation of topical medication, cultures were obtained from the bilateral lower conjunctival sac using a sterile calcium alginate. Contact with eyelids and eyelashes was avoided, and the cultures were immediately inoculated onto 5% sheep blood and chocolate agar plates and sent to the CGMH laboratory. Separate cultures were obtained from both eyes. All culture plates were incubated at 37°C for 3 days and observed. Cultures were deemed positive if colony-forming units were observed. Isolates were identified using standard microbiological protocols. Nasal cultures were taken with swab and transport system and sent for cultures; only *S. aureus* were identified and isolated, if present.

### Bacterial susceptibility testing
To determine susceptibility, the Kirby-Bauer disc diffusion technique was conducted in strict accordance with the guidelines of the National Committee for Clinical Laboratory Standards Institute. Selected antibiotics were tested on different micro-organisms. In addition, an E-test (BioMerieux SA, Marcy-I'Etoile, France) was used to determine the susceptibility to tobramycin and fluoroquinolones including ciprofloxacin, levofloxacin, gatifloxacin and moxifloxacin, which were not included in the antibiotic susceptibility profiles for gram-positive bacteria in our microbiology laboratory. Cefoxitin resistance was considered equivalent to methicillin resistance. Resistance was defined as being 'resistant' or 'intermediate' in susceptibility.

### Data analysis
Descriptive statistics (eg, mean) were calculated for case characteristics. Group comparisons were performed using the $X^2$ test and the Fisher exact test. A p value <0.05 was considered significant.

## RESULTS
### Baseline characteristics
In total, 128 patients who underwent cataract surgery were recruited in the study. The average age of the patients was 68.3 years, and 60 (46.9%) patients were men, 79 (61.7%) had underlying systemic diseases and 53 (41.4%) had at least one of the aforementioned healthcare-associated factors. Postoperatively, all the patients were placed on tobramycin/dexamethasone solution (Alcon, Puurs, Belgium) four times daily. No patient developed endophthalmitis during the follow-up period.

### Bacterial cultures
Fifty-four patients had positive culture results from the conjunctiva. Among them, 14 patients had bilaterally positive culture results. The positive culture rate from

**Table 1** Microbial isolates from the conjunctiva of cataract patients

| Conjunctival isolates | Number (n=84) | Percentage (%) |
|---|---|---|
| Gram-positive bacteria | 76 | 91.7 |
| *Staphylococcus epidermidis* | 14 | 16.7 |
| CNS (other) | 24 | 28.6 |
| *S. aureus* | 2 | 2.4 |
| Gram-positive bacilli* | 29 | 34.5 |
| *Streptococcus* sp | 3 | 3.6 |
| *Enterococcus faecium* | 2 | 2.4 |
| *Micrococcus* sp | 2 | 2.4 |
| Gram-negative bacteria | 7 | 8.3 |
| *Pseudomonas* sp | 4 | 4.8 |
| *Moraxella* species | 1 | 1.2 |
| Other gram-negative species | 2 | 2.4 |
| Fungus | **1** | 1.2 |
| Yeast-like | 1 | 1.2 |

*Gram-positive bacilli included *Lactobacillus* sp, *Corynebacteria* sp, *Listeria* sp and *Bacillus* sp.
CNS, coagulase-negative *staphylococcus*.

**Table 2** Comparison of demographics and characteristics of 26 MSS sp and 14 MRS sp isolated from conjunctiva

| | MSS (n=26) No. | MRS (n=14) No. | p Value* |
|---|---|---|---|
| Gender (men) | 12 | 7 | 0.816 |
| Age (years) | 70.3±10.4 | 71.1±9.1 | 0.218 |
| Healthcare-associated factors | 5 | 4 | 0.694 |
| Systemic history | | | |
| Skin disease | 3 | 1 | >0.999† |
| Liver disease | 2 | 0 | 0.533† |
| Diabetes mellitus | 9 | 5 | >0.999† |
| Heart disease | 5 | 1 | 0.399† |
| Chronic kidney disease | 5 | 0 | 0.143† |
| Hypertension | 16 | 10 | 0.730† |
| Ocular history | | | |
| Ocular surgery | 1 | 1 | >0.999† |
| Personal history | | | |
| Systemic antibiotic history | 0 | 0 | |
| Alcoholism | 1 | 0 | >0.999† |
| Live with kids | 4 | 1 | 0.640† |

*Student's t-test for age comparison, $X^2$ test for others.
†$X^2$ not be a valid test owing to a low number; Fisher exact test was used.
MRS, methicillin-resistant *Staphylococcus*; MSS, methicillin-susceptible *Staphylococcus*.

conjunctiva was 26.6% (68 of 256 eyes). Among the 84 isolates, 76 (91.7%) were gram-positive bacteria, 7 (8.3%) were gram-negative bacteria and 1 (1.2%) was fungus. CNS was the most predominant group of bacterial organisms (38 isolates, 45.2% of the isolates), and more than one-third (14 isolates) was *S. epidermidis*. *S. aureus* isolates were only observed in two eyes, and both were methicillin-sensitive (table 1).

Among *Staphylococcus* sp from conjunctiva, 35% (14 of 40) of staphylococcal isolates were methicillin-resistant. A comparison between the baseline demographic characteristics of patients colonised with methicillin-sensitive and methicillin-resistant *Staphylococcus* sp revealed no significantly associated risk factors between the two groups (table 2).

The positive culture rate from nares for *S. aureus* was 21.9% (28 of 128), and six isolates were methicillin-resistant. None of the study subjects had *S. aureus* colonisation on both conjunctiva and nares.

### Antibiotic susceptibility of *Staphylococcus* species

Table 3 illustrates the antibiotic susceptibility of staphylococcal isolates, including methicillin-resistant and sensitive strains. In total, *Staphylococcus* sp demonstrated the least susceptibility to penicillin (22.5%, 9 of 40), followed by tobramycin (55%, 22 of 40). All staphylococcal isolates were sensitive to vancomycin and teicoplanin. Regarding the fluoroquinolones, the rates of susceptibility to ciprofloxacin, levofloxacin, gatifloxacin and moxifloxacin were 80%, 80%, 75%, and 82.5%, respectively.

In comparison to methicillin-sensitive staphylococcal isolates, methicillin-resistant isolates were significantly less susceptible to sulfamethoxazole–trimethoprim (64.3% vs 96.2%; p=0.014) and tobramycin (28.6% vs 69.2%; p=0.005).

All *S. aureus* isolated from nares were susceptible to sulfamethoxazole–trimethoprim, teicoplanin and vancomycin. MRSA isolates were less susceptible to clindamycin and erythromycin than were methicillin-sensitive *S. aureus* (p=0.001 and 0.007, respectively; table 4).

### DISCUSSION

Although substantial evidence exists regarding the efficacy and safety of intracameral antibiotics, topical antibiotics are still the predominant form of prophylaxis for postoperative endophthalmitis employed by surgeons.[14] Understanding the spectrum of the ocular flora and their antibiotic susceptibility in various geographic locales can assist clinicians in optimising prophylactic antibiotic treatments. We herein present results from a cohort of patients who underwent cataract surgery at our institution. Our findings indicate that CNS is the most common organism isolated from conjunctiva, and that more than one-third of CNS isolates are methicillin-resistant; in Taiwan, methicillin-resistant *Staphylococci* exhibits multidrug resistance

**Table 3** Comparison of antibiotic susceptibilities of 26 MSS sp and 14 MRS sp isolated from conjunctiva

| | Susceptible Strains | | | |
| --- | --- | --- | --- | --- |
| | Total (n=40)N (%) | MSS (n=26)N (%) | MRS (n=14)N (%) | p Value* |
| Clindamycin | 35 (87.5) | 23 (88.5) | 12 (85.7) | >0.999† |
| Erythromycin | 27 (67.5) | 17 (65.4) | 10 (71.4) | >0.999† |
| Penicillin | 9 (22.5) | 9 (34.6) | 0 (0) | 0.016† |
| Sulfamethoxazole–trimethoprim | 34 (85) | 25 (96.2) | 9 (64.3) | 0.014† |
| Teicoplanin | 40 (100) | 26 (100) | 14 (100) | |
| Vancomycin | 40 (100) | 26 (100) | 14 (100) | |
| Ciprofloxacin | 32 (80) | 21 (80.8) | 11 (78.6) | 0.650† |
| Gatifloxacin | 30 (75) | 20 (76.9) | 10 (71.4) | 0.433† |
| Levofloxacin | 32 (80) | 21 (80.8) | 11 (78.6) | 0.650† |
| Moxifloxacin | 33 (82.5) | 22 (84.6) | 11 (78.6) | 0.337† |
| Tobramycin | 22 (55) | 18 (69.2) | 4 (28.6) | 0.005 |

*Student's t-test for age comparison, $X^2$ test for others.
†$X^2$ not be a valid test owing to a low n umber; Fisher exact test was used.
MRS, methicillin-resistant *Staphylococcus;* MSS, methicillin-susceptible *Staphylococcus.*

to sulfamethoxazole–trimethoprim and tobramycin but not to fluoroquinolones.

The culture rate from conjunctiva was 26.6% in this study. To improve culture yield, we inoculated the conjunctival swab samples directly into culture media instead of transport media. Our culture rate was lower than that obtained from other studies[15–18] but moderately higher than in a previous study conducted at CGMH from 2002 to 2008 (18%).[19] By contrast, the culture rate from nares for *S. aureus* was consistent with that of previous studies.[20–23] Insufficient rotation of the cotton swab on the conjunctiva, lengthy shipping time and culture conditions may have contributed to the low culture yield, although the specific cause of the low conjunctival culture rate is uncertain.

Normal conjunctival flora is predominantly composed of gram-positive bacteria and CNS, which were the most commonly isolated organisms in our study. This finding agrees with previous reports wherein CNS represented the most commonly isolated bacteria from ocular surfaces

of patients undergoing ophthalmic surgery[18 24] and from patients with postoperative endophthalmitis.[25 26]

We then examined the antibiotic susceptibility of *Staphylococcus* sp, particularly methicillin-resistant isolates that might exhibit multidrug resistance and be associated with virulent infections.[27] The rate of methicillin resistance among staphylococcal isolates was 35%, in accordance with a reported rate ranging from 35% to 50% in numerous studies since 2000s.[16 18 20 28–32]

Our methicillin-resistant staphylococcal isolates were significantly more resistant to certain antibiotics than their methicillin-sensitive counterparts, including sulfamethoxazole–trimethoprim and tobramycin. The 2005–2006 Ocular TRUST report indicated that MRSA isolates were more resistant to all classes of antibiotics than were MSSA, except for trimethoprim.[6] A recently updated report by the Antibiotic Resistance Monitoring in Ocular Micro-organisms (ARMOR) indicated that methicillin-resistant isolates had a high probability of

**Table 4** Comparison of antibiotic susceptibilities of 22 MSSA and 6 MRSA isolated from nares

| | Susceptible strains | | | |
| --- | --- | --- | --- | --- |
| | Total (n=28)N (%) | MSSA (n=22)N (%) | MRSA (n=6)N (%) | p Value* |
| Clindamycin | 21 (75.0) | 20 (90.9) | 1 (16.7) | 0.001† |
| Erythromycin | 19 (67.9) | 18 (81.8) | 1 (16.7) | 0.007† |
| Penicillin | 2 (7.1) | 2 (9.1) | 0 (0) | >0.999† |
| Linezolid | 28 (100) | 22 (100) | 6 (100) | |
| Sulfamethoxazole–trimethoprim | 28 (100) | 22 (100) | 6 (100) | |
| Teicoplanin | 28 (100) | 22 (100) | 6 (100) | |
| Vancomycin | 28 (100) | 22 (100) | 6 (100) | |
| Tigecycline | 28 (100) | 22 (100) | 6 (100) | |

*Student's t-test for age comparison, $X^2$ test for others.
†$X^2$ not be a valid test owing to a low number; Fisher exact test was used.
MRSA, methicillin-resistant *Staphylococcus aureus;* MSSA, methicillin-susceptible *Staphylococcus aureus.*

concurrent resistance to fluoroquinolones, aminogly-cosides and macrolides.[33] Although reports vary, all the aforementioned studies (including this one) demonstrate that methicillin-resistant organisms exhibit stronger multidrug resistance than their methicillin-sensitive counterparts do.

We paid particular attention to the susceptibility of fluoroquinolones, the most common empiric and prophylactic antibiotics in the ophthalmic field. Growing fluoroquinolone resistance has been widely documented for ocular pathogens and common conjunctival flora.[5 18 20 30 31] In the present study, the susceptibility rate of four tested fluoroquinolones for staphylococcal isolates from conjunctiva ranged from 75% to 82.5%, which is slightly higher than that in earlier reports (50%–80%)%).[16 20 30–32] In contrast to some studies on staphylococcal isolates from normal conjunctival flora and ocular pathogens demonstrating that methicillin resistance can be a marker for fluoroquinolone resistance,[8 11 29 30 33] we did not observe a significant difference in the susceptibility to fluoroquinolones between methicillin-sensitive and methicillin-resistant *Staphylococci*. Taiwan's National Health Insurance Administration reserves fluoroquinolones for the treatment of severe bacterial infections such as corneal ulcers (ie, clinicians may not use fluoroquinolones for prophylactic purposes or mild infections such as conjunctivitis), which might contribute to the relatively high susceptibility to fluoroquinolones in our isolates.

By contrast, the susceptibility rate of staphylococcal isolates to tobramycin in our study was 55%, which is substantially lower than the rates described in recent literature, wherein 90%–100% of staphylococcal isolates were susceptible to tobramycin and gentamicin.[16 18 20 28 31 32] Additionally, methicillin-resistant *staphylococci* were more resistant to tobramycin than their methicillin-sensitive counterparts in our study (71.4% vs 30.8%); the most recent ARMOR update delineated a similar finding, although a lower resistant rate to tobramycin was reported (14% and 2% for methicillin-resistant and methicillin-sensitive CNS, respectively).[33] Tobramycin is the most widely used prophylactic antibiotic for cataracts and other ocular surgeries at our institution and in Taiwan; exposure to subinhibitory concentrations of tobramycin may promote resistance. In sum, *Staphylococci* were more susceptible to fluoroquinolones but less susceptible to tobramycin in our study than in the majority of previous reports.[16 18 20 28 31 32] This may be caused by antibiotic selective pressure (ie, bacteria exposed to certain antibiotics causes emergence of resistant strains). Furthermore, the prevalence of antibiotic use may vary in different regions, which can cause different susceptibility patterns of the isolates. These results highlight the necessity of establishing local epidemiological information.

Our results delineate the tobramycin and multidrug resistance of *Staphylococci* in Taiwan. Tobramycin may not be an effective first-line effort for ophthalmic prophylaxis, particularly for methicillin-resistant *Staphylococci* in Taiwan. Preoperative sterilisation of ocular surface with 5%–10% povidone–iodine is effective in the reduction of bacterial counts on the conjunctiva[34–37] and could reduce the risk of postcataract endophthalmitis without increasing bacterial resistance. In addition to adequate povidone–iodine disinfection, we recommend combining fluoroquinolones with tobramycin as a prophylactic antibiotic, shifting from tobramycin to fluoroquinolones or other modalities such as intracameral injection at the end of surgery to prevent postoperative endophthalmitis.[38 39]

Multiple socioeconomic, environmental and patient factors have been associated with methicillin-resistant organisms in non-ophthalmic literature,[40] however, limited investigation of risk factors has been conducted for patients with ophthalmic problem. If physicians could predict whether a patient undergoing cataract surgery is harbouring methicillin-resistant *Staphylococci*, they can select alternative prophylactic modalities for treatment. However, no differences in possible risk factors between methicillin-resistant and methicillin-sensitive staphylococcal groups were observed in the current study. A recent study of patients undergoing cataract surgery found that age was associated with colonisation by methicillin-resistant organisms, whereas healthcare work was unassociated.[24] Hsu *et al* indicated that antibiotic use within the previous 30 days was a significant risk factor for colonisation by a methicillin-resistant organism,[17] but we excluded the patients who used antibiotics in our study. The variable results in these studies are likely attributable to different sample sizes and different populations. Additional studies are required to establish the risk factors for colonisation by methicillin-resistant organisms.

There are certain limitations to this study. Many of the risk factors were self-reported. Nasal swabs were cultured for *S. aureus* only, because the original study protocol focused on *S. aureus*. Small sample sizes may limit statistical significance. In vitro susceptibility based on serum systemic standards does not always correlate with clinical response because no susceptibility standards are implemented for topical therapies. Finally, because the results of this study are specific to the Taiwanese environment, the findings should not be generalised to other regions or populations.

In conclusion, our study revealed that 35% of staphylococcal isolates from the conjunctiva of patients undergoing cataract were methicillin-resistant. While most *Staphylococci* were susceptible to fluoroquinolones, methicillin-resistant *Staphylococci* were more resistant than their methicillin-sensitive counterparts to tobramycin, the most common prophylactic antibiotic in Taiwan. We hope that clinicians in Taiwan consider the implications of these findings to their current clinical practice.

**Acknowledgements** The authors thank Mr. Lin Yu-Jr in the Biostatistical Center for ClinicalResearch, Chang Gung Memorial Hospital, Taiwan for the assistance in statistical analyses.

**Contributors** CHH: responsible for conception and design of the study. CHHsi and CHH: responsible for provision of patients. YHL and YCK: involved in collection and statistical analysis of data. LYH was involved in writing of the manuscript. YCH and CHH: responsible for critical revision of the manuscript. YCH, CJC, JCS, PHH and CHH: responsible for obtaining funding for the study. All authors: approved the final draft of the manuscript.

**Funding** This work was supported by National Science Council, Taiwan (NSC103-2314-B-182A-043-MY2, NMRPG3D6071) and Chang Gung Memorial Hospital, Taiwan (CMRPG3F0111). The sponsor or funding organization had no role in the design or conduct of this research.

**Competing interests** None declared.

**Ethics approval** The Institutional Review Board of Chang Gung Memorial Hospital.

**Provenance and peer review** Not commissioned; externally peer reviewed.

**Data sharing statement** No additional data are available.

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
