## [Reviewer comments · BMJ Open]

ARTICLE DETAILS

TITLE (PROVISIONAL)	Antibiotic Susceptibility Profiles of Ocular and Nasal Flora in Patients Undergoing Cataract Surgery in Taiwan: An Observational and Cross-sectional Study
AUTHORS	Lin, Yun-Hsuan; Kang, Yu-Chuan; Hou, Chiun-Ho; Huang, Yhu-Chering; Chen, Chih-Jung; Shu, Jwu-Ching; Hsieh, Pang-Hsin; Hsiao, Ching-Hsi

VERSION 1 - REVIEW

REVIEWER	Emily Wong Hong Kong Eye Hospital, Hong Kong
REVIEW RETURNED	31-May-2017

GENERAL COMMENTS	The authors report the prevalence of nasal flora and antibiotic susceptibility of these isolates from patients undergoing elective cataract surgery in a single ophthalmic centre in Taiwan. The following points are suggested to be clarified before publication is considered. A main aim of the study is to identify risk factors for MRSA colonisation / infection. However an important limitation exists in that the authors did not attempt to separate MRSA cases from community associated MRSA (CA-MRSA) which has been demonstrated to be a genetically different phenotype from MRSA exhibiting different infection patterns, genetic makeup and different antibiotic susceptibility. In particular the questionnaire distributed to patients preoperatively focused on risk factors for MRSA but not CA-MRSA. Also it was not mentioned in the design whether cultures were sent for PVL gene testing for CA-MRSA identification. Recent studies (some also published in taiwan) show an increasing prevalence of CA-MRSA especially among the community, hence it is important to address this issue separately. Many of the isolates from this study are known to exist as normal flora, details of the culture positive results are not known for example was the growth heavy/ moderate or commented as commensals only? Were all culture positive cases included? Did any patients exhibit signs of active ocular infection? As many of these isolates are known to be commensals. It would be useful to know if any cases developed postop endophthalmitis in this series and obtain the risk factors for the development of infection in cases with CNS/ MRSA colonisation. The authors did not mention the current practise of antibiotic prophylaxis for cataract surgery in their centre e.g. the dosage and regimen used. Since the prevalence of MRSA colonisation in their
--

	centre seems to be higher than in other international reports, in particular a question that might interest readers is whether the use of intracameral vancomycin prophylaxis is suggested for cases with MRSA colonisation / at risk of MRSA colonisation. The applicability of the prevalence data is also limited from the study in view of the single centre design and the fact that the nasal swabs were only cultured for staph. aureus. It is redundant to mention that methicillin resistant staphylococcus aureus are less sensitive to penicillin when compared to methicillin sensitive staphylococcus aureus. The reported antibiotic sensitivities are based on serum levels and are not particularly relevant to ocular surface infections.
--	--

REVIEWER	Andrzej Grzybowski Dept. of Ophthalmology, University of Warmia and Mazury, Olsztyn, Poland
REVIEW RETURNED	13-Jun-2017

GENERAL COMMENTS	The authors present a prospective study regarding local conjunctival flora and nasal MRSA colonisation. Overall the study appears to be planned, executed and described well. There are however a few areas for improvement. In Methods section, the authors describe the inclusion criteria and a questionnaire that they provided the patients, there is however no mention of exclusion criteria. This is touched on page 16, where authors mention they excluded patients who used antibiotics. I would suggest an explicit description of inclusion and exclusion factors. Both eyes of all participants were tested, and the study describes findings in terms of numbers of isolates. It would be of interest to the reader to describe the data in terms of patients, how many patients had positive culture results, how many had bilaterally positive culture results etc. Page 6, line 128-32 - The statement that "the evaluation of the flora and their drug susceptibility may serve as a surrogate marker for risk of endophthalmitis" is not evidenced and should be excluded. Page 15, line 14-15 authors mention alternatives to tobramycin prophylaxis as topical povidone iodine, it is important to note that topical povidone iodine disinfection is overwhelmingly the standard in prevention of postoperative ophthalmic infections and should not be treated as optional. Moreover, the culture rate from the conjunctiva is unexpectedly low and may suggest that the swabs might have been taken incorrectly as the author noted himself. The authors also compare their results and discuss other studies on this topic published in recent years. It would be also appropriate to highlight the role of pre- and perioperative use of antiseptics as in many cases they have similar effectiveness as antibiotics, but unlike them they do not increase bacterial resistance. Page 15 line 31: the activity of topical antibiotics in prophylaxis of
--

	post-cataract endophthalmitis, esp. preoperative use is based rather on rational thinking than scientific evidence and was shown in several studies to be of low or no effect. Thus, another possible strategy, instead of using a different group of antibiotics, is to rather limit its use. The stable and low rate of endophthalmitis would be another argument for this. In other words, if there was no endophthalmitis increase with present protocol based on tobramycin use (with its relatively low effectiveness) it shows that probably topical antibiotics before surgery are not needed. Table 3. Sub-title mentions MRSA species infections, did the authors mean samples or isolates, as from the study it seems to be colonisations rather than infections.
--	--

REVIEWER	Alessandro Galan San Paolo Ophthalmic Center, San Antonio Hospital, Padova-Italy
REVIEW RETURNED	18-Jun-2017

GENERAL COMMENTS	Its a well written clinical study on the susceptibility profiles of ocular flora.
---

VERSION 1 – AUTHOR RESPONSE

Reviewer 1

1. A main aim of the study is to identify risk factors for MRSA colonisation /infection. However, an important limitation exists in that the authors did not attempt to separate MRSA cases from community associated MRSA (CA-MRSA) which has been demonstrated to be a genetically different phenotype from MRSA exhibiting different infection patterns, genetic makeup and different antibiotic susceptibility. In particular the questionnaire distributed to patients preoperatively focused on risk factors for MRSA but not CA-MRSA. Also it was not mentioned in the design whether cultures were sent for PVL gene testing for CA-MRSA identification. Recent studies (some also published in Taiwan) show an increasing prevalence of CA-MRSA especially among the community, hence it is important to address this issue separately.

Response: Thank you very much for your comments. In our previous study, CA-MRSA played a role in ocular MRSA infections and its annual ratio tended to increase over the 10-year interval (Ophthalmology 2012; 119:522-7.) MRSA could be classified into CA-MRSA and HA-MRSA based on either clinical or molecular criteria. Our questionnaire did include the questions about HA associated factors. We also found the most prominent CA-MRSA clones in Taiwan were ST59/PFGE type C/SCCmec IV/ PVL-negative and ST59/PFGE type D/SCCmec VT/ PVL-positive (Medicine 2015; 94: e1620). The original study protocol focused on *S. aureus*, but we found the most common isolate from conjunctiva was coagulase-negative *Staphylococcus*, and there were only two *S. aureus* isolates, that were methicillin-sensitive. Thus, we did not further analyze these two *S. aureus* isolates. As for coagulase-negative *Staphylococcus*, little literature talked about community onset, and no PVL gene exists (based on previous literature and our unpublished study).

2. Many of the isolates from this study are known to exist as normal flora, details of the culture positive results are not known for example was the growth heavy/ moderate or commented as commensals only? Were all culture positive cases included? Did any patients exhibit signs of active ocular infection?

Response: Thanks for your questions. Cultures were deemed positive if colony forming units were observed (on page 9, lines 4-5); most of the growth was rare and some was light. All culture positive cases were included in the study and no patient presented signs of active ocular infection at the day of surgery. On page 8, lines 4-6, we have added "Exclusion criteria included pregnancy, use of oral or topical antibiotics within 3 months before surgery, and active ocular infection, or conjunctivitis."

3. As many of these isolates are known to be commensals. It would be useful to know if any cases developed postop endophthalmitis in this series and obtain the risk factors for the development of infection in cases with CNS/ MRSA colonisation.

Response: Thank you very much for your suggestions. We have reviewed the medical records of the patients included in the study. No eye developed post-operative endophthalmitis in the follow-up period. Therefore, we are unable to analyze the risk factors for the development of infection in cases with CNS or MRSA colonization. On page 11, line 8, we added "No patient developed endophthalmitis during the follow-up period."

4. The authors did not mention the current practise of antibiotic prophylaxis for cataract surgery in their centre e.g. the dosage and regimen used. Since the prevalence of MRSA colonisation in their centre seems to be higher than in other international reports, in particular a question that might interest readers is whether the use of intracameral vancomycin prophylaxis is suggested for cases with MRSA colonisation / at risk of MRSA colonisation.

Response: Thanks for your comments. We have added the current practice of antibiotic prophylaxis for cataract surgery in our center. On page 11, lines 6-7, "Postoperatively, all the patients were placed on tobramycin/dexamethasone solution (Alcon, Puurs, Belgium) 4 times daily."

The susceptibility rate of staphylococcal isolates to tobramycin in our study was 55%, which is substantially lower than the rates described in recent literature. However, the susceptibility rate of fluoroquinolones for staphylococcal isolates from conjunctiva in Taiwan is slightly higher than that in earlier reports, and all staphylococcal isolates were susceptible to vancomycin. Thus, from page 16, lines 18-20 to page 17, lines 1-2, "We recommend combining fluoroquinolones with tobramycin as a prophylactic antibiotic, shifting from tobramycin to fluoroquinolones or other modalities such as intracameral injection at the end of surgery to prevent postoperative endophthalmitis."

5. The applicability of the prevalence data is also limited from the study in view of the single centre design and the fact that the nasal swabs were only cultured for staph. aureus.

Response: Yes, they are limitations of this study, so we had pointed them out in the original manuscript (from page 17, line 18 to page 18, line 3), "There are certain limitations to this study. Many of the risk factors were self-reported. Nasal swabs were cultured for *S. aureus* only, because the original study protocol focused on *S. aureus*. Small sample sizes may limit statistical significance. In vitro susceptibility based on serum systemic standards does not always correlate with clinical response because no susceptibility standards are implemented for topical therapies. Finally, because the results of this study are specific to the Taiwanese environment, the findings should not be generalized to other regions or populations."

6. It is redundant to mention that methicillin resistant staphylococcus aureus are less sensitive to penicillin when compared to methicillin sensitive staphylococcus aureus.

Response: Thank you for the suggestion, we have deleted the description on page 13, line 10.

7. The reported antibiotic sensitivities are based on serum levels and are not particularly relevant to ocular surface infections.

Response: Yes, it is the common limitation of such studies, so we had described it in the limitation section of the original manuscript, on page 18, lines 1-3, "In vitro susceptibility based on serum systemic standards does not always correlate with clinical response because no susceptibility standards are implemented for topical therapies."

Reviewer: 2

The authors present a prospective study regarding local conjunctival flora and nasal MRSA colonisation. Overall the study appears to be planned, executed and described well. There are however a few areas for improvement.

1. In Methods section, the authors describe the inclusion criteria and a questionnaire that they provided the patients, there is however no mention of exclusion criteria. This is touched on page 16, where authors mention they excluded patients who used antibiotics. I would suggest an explicit description of inclusion and exclusion factors.

Response: Thank you very much for your suggestions. We have added the exclusion criteria. On page 8, lines 4-6, "Exclusion criteria included pregnancy, use of oral or topical antibiotics within 3 months before surgery, and active ocular infection, or conjunctivitis."

2. Both eyes of all participants were tested, and the study describes findings in terms of numbers of isolates. It would be of interest to the reader to describe the data in terms of patients, how many patients had positive culture results, how many had bilaterally positive culture results etc.

Response: Thank you very much for your comments. On page 11, lines 10-11, we have added "Fifty-four patients had positive culture results from the conjunctiva. Among them, 14 patients had bilaterally positive culture results."

3. Page 6, line 128-32 - The statement that "the evaluation of the flora and their drug susceptibility may serve as a surrogate marker for risk of endophthalmitis" is not evidenced and should be excluded.

Response: Thank you for the suggestion. We have deleted the statement.

4. Page 15, line 14-15 authors mention alternatives to tobramycin prophylaxis as topical povidone iodine, it is important to note that topical povidone iodine disinfection is overwhelmingly the standard in prevention of postoperative ophthalmic infections and should not be treated as optional.

Response: Thank you very much for your great comments. Yes, topical povidone iodine disinfection is the standard in prevention of postoperative ocular infections. We have modified our recommendations as "In addition to adequate povidone-iodine disinfection, we recommend combining fluoroquinolones with tobramycin as a prophylactic antibiotic, shifting from tobramycin to fluoroquinolones or other modalities such as intracameral injection at the end of surgery to prevent postoperative endophthalmitis." from page 16, line 18 to page 17, line 2.

5. Moreover, the culture rate from the conjunctiva is unexpectedly low and may suggest that the swabs might have been taken incorrectly as the author noted himself.

Response: Yes, we tried our best, but did not know the real cause. We had discussed it on page 13, lines 12-20, "The culture rate from conjunctiva was 26.6% in this study. To improve culture yield, we inoculated the conjunctival swab samples directly into culture media instead of transport media. Our culture rate was lower than that obtained from other studies but moderately higher than in a previous study conducted at CGMH from 2002-2008 (18%).¹⁹ By contrast, the culture rate from nares for S.

aureus was consistent with that of previous studies. Insufficient rotation of the cotton swab on the conjunctiva, lengthy shipping time, and culture conditions may have contributed to the low culture yield, although the specific cause of the low conjunctival culture rate is uncertain.”

6. The authors also compare their results and discuss other studies on this topic published in recent years. It would be also appropriate to highlight the role of pre- and perioperative use of antiseptics as in many cases they have similar effectiveness as antibiotics, but unlike them they do not increase bacterial resistance.

Response: Thank you very much for your suggestion. We highlighted the importance of ocular surface disinfection. On page 16, lines 16-8, “Preoperative sterilization of ocular surface with 5-10% povidone-iodine is effective in reduction of bacterial counts on the conjunctiva, and could reduce the risk of post-cataract endophthalmitis without increasing bacterial resistance.”

7. Page 15 line 31: the activity of topical antibiotics in prophylaxis of post-cataract endophthalmitis, esp. preoperative use is based rather on rational thinking than scientific evidence and was shown in several studies to be of low or no effect. Thus, another possible strategy, instead of using a different group of antibiotics, is to rather limit its use. The stable and low rate of endophthalmitis would be another argument for this. In other words, if there was no endophthalmitis increase with present protocol based on tobramycin use (with its relatively low effectiveness) it shows that probably topical antibiotics before surgery are not needed.

Response: Thank you very much for your comments. In our institution, prophylactic antibiotics were used at the end of cataract surgery (on page 11, lines 6-7) rather than preoperatively. Indeed, the best way to avoid antibiotic resistance is to limit its use. However, we may need analyze our data of postoperative endophthalmitis to get a conclusion, but such information is not available currently. Although your viewpoint is so great, we decided to keep our recommendation about postoperative antibiotic use in Taiwan. Sorry!

8. Table 3. Sub-title mentions MRSA species infections, did the authors mean samples or isolates, as from the study it seems to be colonisations rather than infections.

Response: Thanks for pointing out our mistake. We have changed the titles of Tables 2 and 3.

VERSION 2 – REVIEW

REVIEWER	Wong Emily Suhan Hong Kong Eye Hospital Hong Kong, China
REVIEW RETURNED	29-Jun-2017

GENERAL COMMENTS	This cross-sectional study describes prevalence data on conjunctival and nasal flora and their antibiotic susceptibility profiles from patients undergoing cataract surgery in Taiwan. The study outcomes are better described after revision, authors have also provided suggestions regarding antibiotic prophylaxis in their locality which is of clinical relevance.
--

REVIEWER	Andrzej Grzybowski Dept. of Ophthalmology, University of Warmia and Mazury, Olsztyn, Poland
REVIEW RETURNED	04-Jul-2017

GENERAL COMMENTS

Authors revised the ms according to reviewers suggestions and now it is suitable for publication.